A transformer-based framework for enterprise sales forecasting

Sun Yupeng 1 syupeng@126.com
Li Tian 2
1 School of Accounting, Yunnan University of Finance and Economics , Yunnan , China
2 School of Accounting, Tianjin University of Commerce , Tianjin , China
Nguyen Hoang
Electronic publication date: 2024 Nov 21
Publication date: 2024
Volume: 10
Electronic Location ID: e2503
Received 2024 Aug 14; Accepted 2024 Oct 21
Copyright: © 2024 Sun and Li
Copyright year: 2024
Copyright holder: Sun and Li
License: This is an open access article distributed under the terms of the Creative Commons Attribution License, which permits unrestricted use, distribution, reproduction and adaptation in any medium and for any purpose provided that it is properly attributed. For attribution, the original author(s), title, publication source (PeerJ Computer Science) and either DOI or URL of the article must be cited.
License URL: https://creativecommons.org/licenses/by/4.0/

Keywords: Sales forecasting, Transformers, Deep learning, Business intelligence

Funding: National Social Science Foundation of China 22BGL218 This work was supported by the National Social Science Foundation of China (Project No. 22BGL218), “On the Bubble Risk Identification and Prevention of Local Governments’ Competition in Green Innovation under the Goal of Carbon Neutrality.” The funders had no role in study design, data collection and analysis, decision to publish, or preparation of the manuscript.

==============================
Sales forecasting plays an important role in business operations as it impacts decisions on inventory management, allocation of resources, and financial planning. Accurate sales predictions are essential for optimizing cash flow management, adapting marketing and sales strategies, and facilitating strategic planning. This study presents a computational framework for predicting business sales using transformers, which are considered one of the most powerful deep learning architectures. The design of our model is specifically tailored to accommodate tabular data with low dimensions. The experimental results demonstrated that our proposed method surpasses conventional machine learning models, achieving reduced mean absolute error (MAE), mean square error (MSE), and root mean square error (RMSE), as well as higher R2 values of nearly 0.95. The results confirmed that the model is applicable not only to this research but also to similar studies that use low-dimensional tabular data. The improved accuracy and stability of our model demonstrate its potential as a useful tool for enhancing sales prediction, therefore facilitating more informed decision-making and strategic planning in corporate operations.

Introduction

Background

Sales forecasting plays a vital role in business activities from profit estimation to decision-making (Lawrence, O’Connor & Edmundson, 2000; Wacker & Lummus, 2002). Effective sales forecasting shapes decisions related to inventory control (Fildes & Beard, 1992; Snyder, Koehler & Ord, 2002; Snyder, 2002), resource allocation (Wacker & Lummus, 2002; Gupta & Kohli, 2006; Sugiarto, Sarno & Sunaryono, 2016), and financial planning (Osadchiy, Gaur & Seshadri, 2013; Zhu, 2023; Xu, Li & Donta, 2024). Sales history data, in combination with socioeconomic factors such as current market dynamics, economic indicators, consumer trends, and seasonal impacts, are typically used to predict sales for upcoming periods (Gao et al., 2014; Cheriyan et al., 2018). Effective forecasts are crucial for optimizing cash flow management (Dharan, 1987) and adjusting marketing (Biswas, Sanyal & Mukherjee, 2023) and sales strategies (Janczura & Michalak, 2020; Zhu, Bahadur & Ali, 2023). An increase in demand might lead to ramped-up production or enhanced promotional activities to maximize sales, while accurate forecasting of a sales decline might prompt cost-cutting measures (Mccarthy et al., 2006). Furthermore, sales forecasting also serves as a cornerstone of strategic planning, helping to set achievable sales goals that synchronize the efforts of marketing, sales, production, and logistics teams (Davis & Mentzer, 2007). Improved forecasting accuracy enhances customer satisfaction by ensuring that businesses can meet demand promptly and reliably (Lyu & Choi, 2020; He, 2022; Ban et al., 2023). Therefore, improving sales forecasting techniques is one of the major approaches to maintain a business’s operational efficiency and financial stability, allowing it to adapt quickly to changing market conditions and consumer preferences, thereby sustaining growth and competitiveness (Lawrence & O’Connor, 2000).

In the era of artificial intelligence (AI) (Sohrabpour et al., 2021) and big data (Thomassey & Zeng, 2018), sales forecasting has undergone a transformative shift, harnessing advanced technologies to predict future sales with unprecedented precision. AI algorithms and machine learning (ML) models are now central to analyzing vast amounts of data, identifying patterns and trends that were previously indiscernible (Chen & Lu, 2016). This has allowed businesses to refine their forecasts beyond traditional statistical modeling methods, integrating real-time data from a variety of sources including market shifts, consumer behavior online, social media sentiments, and even weather patterns (Ramos, Santos & Rebelo, 2015; Leow, Nguyen & Chua, 2021). The integration of big data analytics into sales forecasting provides a holistic view of the marketplace (Boone et al., 2019). Companies can now process and analyze data from diverse and extensive datasets instantaneously. This ability enables more dynamic and agile responses to market changes, as AI can quickly adapt forecasts in response to promotional campaigns or sudden shifts in consumer demand, providing businesses with the agility to optimize inventory levels and tailor marketing strategies effectively (Forrest & Hoanca, 2015). Moreover, AI-driven tools offer predictive insights with a level of granularity that allows companies to segment customers more precisely, tailor products to specific markets, and adjust prices dynamically (Chen et al., 2024). The predictive power of AI also extends to identifying potential market opportunities and risks before they fully emerge, giving companies strategic advantages. Overall, the advent of AI and big data has not only enhanced the accuracy of sales forecasts but has also revolutionized how businesses strategize and operate (Thomassey & Zeng, 2018; Sohrabpour et al., 2021). As these technologies continue to evolve, they promise to unlock even deeper insights and drive smarter, data-driven decision-making across industries. This advancement is important for businesses aiming to remain competitive in a rapidly changing economic landscape (Weber & Schütte, 2019).

Related works

For decades, numerous computational frameworks employing machine learning and deep learning have been developed for sales forecasting (or prediction). Chu & Zhang (2003) conducted a comparative study to compare the performance of linear and nonlinear models for retail sales forecasting. Das & Chaudhury (2006) developed a model to estimate the sales fluctuations of a footwear company over a period of time using recurrent neural networks. Their model was designed to predict weekly retail sales to minimize the uncertainty in short-term sales planning. Ni & Fan (2011) proposed a forecasting model for the fashion retail using real-time data, combining two types of prediction: long-term and short-term. Beheshti-Kashi et al. (2014) implemented various state-of-the-art methods to construct forecasting models for fashion and new products under different modeling strategies. Kaneko & Yada (2016) utilized simple deep neural networks to develop a prediction model for retail store sales. Ribeiro, Seruca & Durão (2017) created a model specifically for predicting sales in a pharmaceutical distribution firm using exponential smoothing time-series. Their research tackled two major issues: exploring inventory allocation strategies to prevent stock shortages and forecasting sales to maintain sufficient levels of medicine inventory. Punam, Pamula & Jain (2018) introduced a two-level statistical model for forecasting big mart sales using linear regression, support vector regression, and cubist. Tsoumakas (2018) investigated a series of machine learning techniques used for food sales prediction. Kohli, Godwin & Urolagin (2020) used linear regression and k-nearest neighbors regression to develop sales prediction models. Yao (2023) employed three machine learning algorithms, including decision tree, random forest, and k-neighbors regression, to build a forecasting model for Walmart sales prediction.

Motivations

Since most deep learning architectures used for sales forecasting are relatively simple, the potential of deep learning, especially more advanced models, has not been fully exploited. In our study, we developed a computational framework for enterprise sales forecasting using a streamlined transformer for low-dimensional tabular data. Our model draws inspiration from the TabTransformer (Huang et al., 2020), with minor modifications. As a transformer-based architecture specifically designed for tabular data, it performs more effectively on this type of data compared to traditional neural networks. Furthermore, all transformer-based architectures are characterized by attention mechanisms, which have been recognized as exceptionally powerful in recent years (Vaswani et al., 2017; Huang et al., 2020; Badaro, Saeed & Papotti, 2023; Mao, 2024). Transformers and their variants have been widely used in natural language processing. After training, our developed models are rigorously assessed and compared against multiple conventional machine learning and deep learning algorithms. Additionally, model stability is evaluated by repeating experiments across multiple random trials.

Proposed model

Model architecture

Figure 1 visualizes the model architecture proposed in our study. The model receives two types of input features: categorical and numerical features. The categorical features are first passed through a Column Embedding layer (see subsection Column Embedding) before entering a Multi-head Attention layer. The embedding vectors are loaded into a Multi-head Attention layer, summed with the embedding residual, and normalized. The summed attention outputs continue to enter a Feed Forward layer. The Feed Forward layer’s outputs are then summed with the attention residual and normalized to create categorical attention outputs. The numerical features are normalized before being processed by the three Feed Forward layers. The Feed Forward layers’ outputs are used as Query (Q), Key (K), and Value (V) vectors for the Self-Attention layer. The Self-Attention layer learns numerical features via Q, K, and V vectors to create numerical attention outputs. The categorical and numerical attention outputs are concatenated and then transferred to a Multi-layer Perceptron (MLP) block to predict final outcomes. Multi-layer Perceptron block is specified by three linear layers. The input size of these linear layers is 107, 428, and 214, respectively. The output size of the prior linear layers is equal to the input size of the posterior layers. The Rectified Linear Unit (ReLU) is used as the activation function. In the TabTransformer architecture, Huang et al. (2020) applied one layer of normalization for the numerical features branch. In contrast, we added an attention layer after the normalization layer for these features.

Figure 1 Model architecture proposed in our study.

The network is designed with two major input blocks: Column Embedding and Normalization, for processing categorical (discrete or string) and numerical (continuous) features, respectively. Both types of features are passed through distinct attention layers for selective learning of important features. The selected attention features are then concatenated before being transferred to the Multilayer Perceptron block.

Column embedding

Column embedding is a technique specifically designed for modeling tabular data, utilizing the strengths of transformer models to address the unique challenges of categorical and numerical features in these datasets (Huang et al., 2020). In this approach, categorical features (columns) are transformed into dense vector representations, or embeddings, which allow the model to effectively learn from them. These embeddings capture both the identity and the contextual relationships between categories, which is crucial for understanding feature interactions that heavily influence the target variable (Mao, 2024). Numerical features are also incorporated alongside categorical embeddings, enabling the model to leverage both types of data in its predictions. The embedding process is typically achieved through learned embeddings, where each category is mapped to a vector in a continuous space. This allows the model to identify hidden relationships between categories that may not be obvious in the raw data. Furthermore, the contextual embeddings learned by the model are resilient to missing and noisy data, ensuring robust performance in real-world scenarios where data quality can vary. By integrating column embeddings, the model consistently outperforms traditional deep learning methods for tabular data, delivering higher predictive accuracy and improved generalization across a wide range of datasets (Huang et al., 2020; Mao, 2024).

Self-attention

The self-attention mechanism, derived from the transformer architecture, calculates attention scores to determine the relevance of each feature in relation to the others in the input data. This enables the model to concentrate on critical feature interactions that are essential for accurate predictions. Given an input matrix of size X∈Rn×d, n represents the number of features and d is the dimension of the embedding. The attention mechanism can be computed based on three fundamental components: Query (Q), Key (K), and Value (V). The Q, K, and V matrices are calculated through learnable weight matrices WQ, WK, and WV∈Rd×dk, respectively where dk is the key dimension. The attention scores are computed by taking the dot product between the Q and K matrices, scaled by the square root of the key dimension dk, as follow:

(1) Attention(Q,K,V)=softmax(QKTdk).

The dot product QKT calculates the relevance between each query and key, while scaling by dk helps maintain stable gradients when dk is large. The softmax function then converts the attention scores into probabilities for better interpretation.

Dataset description

In our study, we utilized the dataset from the Kaggle competition “Walmart Recruiting- Store Sales Forecasting” (Kaggle, 2014). This dataset consists of three dataframes, which we merged to form a single dataframe containing 14 variables. These include 11 continuous variables: Size, Temperature, Fuel_Price, CPI, Unemployment, Total_MarkDown, max, min, mean, median, and std; along with three categorical variables: Store, Dept, and Type. The Weekly_Sales variable is labels for prediction. Since the dataset contains time-series samples, we created three sets of data: training, validation, and test sets based on the time order. The training set contains 331,742 samples from 05/02/2010 to 30/03/2012, equivalent to 113 weeks. The validation and test sets have 44,192 and 44,278 samples, respectively, both equivalent to 15 weeks. The validation data were recorded from 06/04/2012 to 13/07/2012, while the test data were recorded from 20/07/2012 to 26/10/2012. The whole dataset contains data recorded from 05/02/2010 to 26/10/2012, equivalent to 143 weeks with 420,212 samples. Table 1 summarizes information on datasets used for model training, validation, and testing. The data preprocessing was executed using module preprocessing, scikit-learn library (Pedregosa et al., 2011) version 1.4.1 in Python 3.11.8 environment.

Table 1 Information on datasets used for model training, validation, and testing.

Dataset	Time	Number of samples	Number of weeks	
Training	05/02/2010 to 30/03/2012	331,742	113	
Validation	06/04/2012 to 13/07/2012	44,192	15	
Test	20/07/2012 to 26/10/2012	44,278	15	
Total	05/02/2010 to 26/10/2012	420,212	143	

Experiments

Overview of modeling strategy

Unlike normal datasets whose samples can be randomly split, the time-series dataset contains ordered samples which cannot be reversed. Therefore, the training process was designed to direct all models to learn data sequentially. The initial training set contained samples of 113 weeks, and the validation set contained samples of 15 weeks, which were used to develop all models, including ours. Figure 2 visualizes the modeling strategy in our study. This strategy includes two phases: (i) training and validation and (ii) training and testing.

Figure 2 Modeling strategy in our study.

In iteration 1, the models were trained with data corresponding to 113 weeks and validated by data from week 1 (in the next 15 weeks). In iteration 2, the week-1 validation data were accumulated in the initial training set for updating the models by re-training them with the updated training set. This process was iteratively repeated until the models were validated by the data from week 15. During the training and validation phase, the models were retrained 15 times.

After completing the training and validation phase, we conducted the training and evaluation phase. In the second phase, the models were also created, updated, and evaluated in the same manner as the first phase. Since the test set contains data from the last 15 weeks, the models in the training and evaluation phase were also updated 15 times. All validation and evaluation results were recorded for further analysis.

Training our model

Our model was optimized using the Adam optimizer. The time required to complete one training epoch varies from 200 to 250 s. In the training and validation phase, the first model (at iteration 1) was trained over 30 epochs with a learning rate of 0.001. The next updated models were trained with an additional 1-5 epochs. The training process was terminated when the validation loss reached the bottom and started to rise. In the training and evaluation phase, the first model was retrained with a secondary training set including all samples of training and validation data. After the first models were obtained, the next updated models were retrained with one additional epoch and evaluated by the data of the next week. The training process of the second phase was stopped after 15 iterations.

Training machine learning models

To perform comparative analysis, we trained four other models using conventional machine learning algorithms, including k-nearest neighbor ( k-NN), linear regression (LR), random forest (RF), and eXtreme gradient boosting (XGB). Two deep learning models based on gated recurrent unit (GRU) (Elsworth & Güttel, 2020), and long short-term memory (LSTM) (Velarde et al., 2022) models were also implemented for compassion. A deep learning model based on TabTransformer (Huang et al., 2020) were trained to serve as a baseline to compare with our model. Each method leverages a distinct principle of learning from data. k-nearest neighbor works on the principle of proximity, predicting the label of a query point based on the most frequent label (classification) or average value (regression) of its k closest neighbors in the feature space (Nguyen, Tay & Chui, 2015). Linear regression works by fitting a linear equation to observed data, establishing a relationship between one dependent variable and one or more independent variables to predict outcomes. Random forest improves on the decision tree method by creating an ensemble of trees where each tree is trained on a subset of the data and features, thus reducing variance and avoiding overfitting (Nguyen-Vo et al., 2019). eXtreme gradient boosting enhances traditional gradient boosting by optimizing the algorithm’s speed and efficiency, using a more regularized model formalization to control over-fitting, which makes it robust and highly accurate even on large and complex datasets (Pham et al., 2019). Gated recurrent unit and long short-term memory are two effective and commonly used recurrent neural networks for learning sequential data (Le et al., 2019).

Assessment metrics

To evaluate the performance of all models, we used mean absolute error (MAE), mean square error (MSE), root mean square error (RMSE), and coefficient of determination (R2), and mean absolute percentage error (MAPE). The mathematical formula of these metrics are expressed as:

(2) MAE=1N∑i=1N|yi−y^i|,

(3) MSE=1N∑i=1N(yi−y^i)2,

(4) RMSE=1N∑i=1N(yi−y^i)2,

(5) R2=∑i=1N(y^i−x¯)2∑i=1N(yi−x¯)2,

(6) MAPE=1N∑i=1N|yi−y^iyi|,

where N is the total number of samples, y^i is a predicted value, yi is a ground truth value, and x¯ is a mean value over the total samples. These metrics are commonly used to compute the prediction error of regression models.

Computing platform

In our study, all modeling experiments were conducted on a personal computer (AMD Ryzen 7 5800X 8-Core Processor 3.80 GHz and 4 ×8 GB RAM). The deep learning and machine learning models were implemented using PyTorch (Paszke et al., 2019) version 2.0.0 (CUDA Toolkit 11.7) and scikit-learn library (Pedregosa et al., 2011) version 1.4.1 in Python 3.11.8 environment.

Results and discussion

Model robustness

Table 2 summarizes the comparative analysis between our model and other conventional machine learning models over 15 iterations to assess the stability of all implemented models. The results show that our model obtains smaller values of MSE, RMSE, MAE, and MAPE and higher R2 values compared to other machine learning models. In terms of MAE, our model is the best-performing, followed by the random forest, eXtreme gradient boosting, k-nearest neighbor, and linear regression models. The MSE, RMSE, and R2 values of all models also follow the same ranking order. These findings reveal that our proposed model works more effectively than other machine learning models often employed to address similar problems. Additionally, the adaptivity of the model in processing and learning tabular data is observed. Our results indicate that the variation of all recorded measures is very small, indicating highly stable performance regardless of sampling. The means of the MAE, MSE, and RMSE values are 3.12, 25.76, and 5.08, respectively, while the R2 value is 0.9462. Figure 3 shows the predicted values plotted against actual observations, indicating that the model performs more effectively in predicting lower values, while its predictive performance tends to decline for higher values. This is due to the fact that the dataset has more samples with lower values.

Table 2 Performance of our model compared to other machine learning models.

Model	MSE	RMSE	MAE	MAPE	R2	
k-nearest neighbor	213.71 ± 4.65	14.62 ± 1.54	8.68 ± 0.56	72.53 ± 2.35	0.55 ± 0.02	
Linear regression	434.88 ± 10.12	20.85 ± 1.81	14.64 ± 0.87	166.71 ± 3.36	0.09 ± 0.01	
Random forest	64.56 ± 2.11	8.03 ± 1.24	5.27 ± 0.54	60.99 ± 2.17	0.87 ± 0.03	
eXtreme gradient boosting	239.44 ± 3.59	15.47 ± 1.67	9.61 ± 0.75	102.68 ± 2.95	0.5 ± 0.02	
GRU-based model	37.84 ± 3.23	6.15 ± 1.14	3.76 ± 0.41	16.08 ± 1.25	0.92 ± 0.09	
LSTM-based model	34.64 ± 4.68	5.89 ± 1.39	3.56 ± 0.48	30.26 ± 1.82	0.93 ± 0.08	
TabTransformer	85.72 ± 3.74	9.26 ± 1.37	6.21 ± 0.59	79.89 ± 2.26	0.82 ± 0.09	
Ours	25.76 ± 2.47	5.08 ± 1.23	3.12 ± 0.37	9.49 ± 1.11	0.95 ± 0.08	

Figure 3 Predicted values plotted against actual observations.

Limitations

Despite the promising results, our work also has limitations that need to be addressed to improve the robustness and applicability of our model. Since the model performance was evaluated on a specific medium-sized dataset, the proposed method may not fully capture the diversity and complexity of real-world problems, limiting the applicability of our findings to similar topics. The comparative analysis is restricted to benchmarking our work against a small number of conventional machine learning models. In the future, more advanced models, including data-centric AI methods (Wang et al., 2024), can be implemented to further assess the model’s performance.

Conclusion

In our study, we proposed a computational framework for enterprise sales forecasting using transformers, one of the most effective deep learning architectures. Our model was carefully developed to adapt to low-dimensional tabular data. Our findings suggest that our proposed method is highly robust and reproducible with low variation. Compared to other machine learning methods, our model showed better performance with lower MAE, MSE, and RMSE, and higher R2 values of nearly 0.95. These results not only confirm the applicability of the model in this research but also its potential in similar studies using low-dimensional tabular data.

Supplemental Information

Supplemental Information 1 The Python code and raw data.

Additional Information and Declarations

Competing Interests

Author Contributions

Data Availability

The authors declare that they have no competing interests.

Yupeng Sun conceived and designed the experiments, performed the experiments, analyzed the data, performed the computation work, prepared figures and/or tables, authored or reviewed drafts of the article, and approved the final draft.

Tian Li conceived and designed the experiments, performed the experiments, analyzed the data, prepared figures and/or tables, authored or reviewed drafts of the article, and approved the final draft.

The following information was supplied regarding data availability:

The code and data used in the experiments are available in the Supplemental File.

The data is originally from Kaggle competition “Walmart Recruiting-Store Sales Forecasting” available at: https://www.kaggle.com/c/walmart-recruiting-store-sales-forecasting.

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
