# Peer review of "A transformer-based framework for enterprise sales forecasting"

_PeerJ Computer Science, doi:10.7717/peerj-cs.2503_

## Round 0.1 · original submission · Major Revisions

Based on reviewers' comments, we have found that your work needs to be significantly improved. However, we are happy to offer you an opportunity to revise your work. Please check and address all points raised by reviewers.

Reviewer 1 ·

Basic reporting

The authors proposed a transformer-based framework for enterprise sales forecasting, claiming it effectively addresses the challenges of sales forecasting with improved performance. The language is clear, and the manuscript is well-structured with informative sections. However, there are several issues that need to be addressed, as listed below.

Experimental design

+ The proposed model includes a self-attention mechanism, but there is no detailed description of how the attention layer operates. Additionally, the formula for computing attention scores should be included.
+ In the "Training Machine Learning Model" section, the authors mention using conventional machine learning algorithms for prediction models, which were then used for benchmarking. However, this comparison might not be fair, as these algorithms may not perform well with time-series data. I recommend conducting more experiments with models that are better suited for sequential data, such as GRU or LSTM. While this section can serve as a baseline comparison, it should not be used to demonstrate the effectiveness of your method.
+ In the "Assessment Metrics" section, I suggest computing additional metrics, such as MAPE, for a more comprehensive evaluation.
+ In the "Model Reproducibility" subsection, your models appear to significantly outperform others, suggesting that your method might be overqualified compared to these methods. I recommend revising this section after revising the "Training Machine Learning Model" section.
+ It should be better to include a plot comparing predicted values with ground truth values to visually explain your evaluation to readers.

Validity of the findings

Statistical evidence is needed to support your results. If the results presented are average values, please show them as mean ± standard deviation. If not, please review and revise accordingly.

Cite this review as

·

Basic reporting

This work focuses on using transformer architecture to develop a time-series model for predicting enterprise sales. The proposed architecture demonstrates its ability to handle both categorical and numerical features, enhancing attention-based learning on data features. The manuscript is well-organised, and the writing is professional and comprehensible.

Experimental design

Although the work is interesting and has merits, some minor points need revision to enhance its quality:

- A section explaining the concept of attention for newcomers is necessary.
- Why are numerical features processed by the self-attention layer while categorical features are handled by multi-head attention?
- How do these approaches differ?
- What is Column Embedding? It should be introduced before presenting it in the architecture.
- Is there a reason for selecting only traditional machine learning models for comparative analysis? These models might not perform well with large datasets, particularly time-series data. Consider implementing recurrent neural networks or other models to provide more persuasive evidence of how well your model performs.
- The number "2" in "R²" in Table 2 should be superscripted.

Validity of the findings

- In lines 192-193: “These results provide strong statistical evidence to support the stability of our methods for further implementation.” I don’t see any evidence here. Could you please clarify? It would be helpful to present results with standard deviation or a 95% confidence interval, etc.

Cite this review as

Reviewer 3 ·

Basic reporting

- The paper is easy to understand and includes necessary components.
- However, the introduction need to revise to introduce clearly the subject and motivation.
- The authors should consider move the Motivation section to Introduction section for better message delivery.

Experimental design

- Figure 1 and the Proposed Model need to mention what are categorial and numerical features, how to extract or preprocess them?
- The authors should state the difference between the proposed method with TabTransformer, what are the modifications?
- Lack of ablation studies to validate the modifications.

Validity of the findings

- The authors should evaluate their proposed method on newer dataset because the current dataset is too old which is hard to reflect the current situation of economy.
- In addition, the authors should evaluate with more datasets instead of only 1 dataset to show the effectiveness and generalization of the proposed method.
- Comparing Deep learning-based method with traditional machine learning methods seems unfair.
- The authors based on TabTransformer to propose their method, then, the authors should compare with TabTransformer (as a baseline).

Additional comments

- Line 90-92 should add more references to support the claim "all architectures" but cite only 1 paper and it is pretty outdated (2017).

Cite this review as

---

## Round 0.2 · accepted · Accept

Thank you for your revised "A transformer-based framework for enterprise sales forecasting" work in PeerJ Computer Science. Authors have shown great effort in addressing reviewer's concerns about this work.

Reviewer 1 ·

Basic reporting

The revised version of the manuscript meets the journal's standards. I have no more comment to add.

Experimental design

The authors have revised the manuscript, including performing extra experiments, to address our comments. I have no more comments to add.

Validity of the findings

The experiments and evaluations performed satisfactorily. The discussion has been improved. This includes discussion about limitations of the method. I have no more comments to add.

Additional comments

No comment.

Cite this review as

·

Basic reporting

- The revised manuscript is well-written and explains complex concepts effectively, making it easy to follow.
- This version is well-organized, and the literature is nicely referenced.
- The manuscript has addressed all previous concerns and now ready for publication at the journal.

Experimental design

- The experimental design has been improved by adding GRU and LSTM models for a more comprehensive comparison to traditional machine learning models.
- The explanation of categorical and numerical features through different attention mechanisms is well-explained.
- The authors have provided ample details on data preprocessing, model architecture, and evaluation metrics.
- The experimental design is sufficient and aligned with the study's scope.

Validity of the findings

- The statistical reporting looks good after including standard deviations.
- The proposed transformer-based model demonstrates clear performance superiority compared to baseline models, showing high stability and accuracy.
- While the study focuses on a specific dataset, the results strongly support the authors' claims about the model's effectiveness in handling tabular data.

Cite this review as